# Mirror Gradient: Towards Robust Multimodal Recommender Systems via Exploring Flat Local Minima

## ABSTRACT

Multimodal recommender systems utilize various types of information to model user preferences and item features, helping users discover items aligned with their interests. The integration of multimodal information mitigates the inherent challenges in recommender systems, e.g., the data sparsity problem and cold-start issues. However, it simultaneously magnifies certain risks from multimodal information inputs, such as information adjustment risk and inherent noise risk. These risks pose crucial challenges to the robustness of recommendation models. In this paper, we analyze multimodal recommender systems from the novel perspective of *flat local minima* and propose a concise yet effective gradient strategy called Mirror Gradient (MG). This strategy can implicitly enhance the model's robustness during the optimization process, mitigating instability risks arising from multimodal information inputs. We also provide strong theoretical evidence and conduct extensive empirical experiments to show the superiority of MG across various multimodal recommendation models and benchmarks. Furthermore, we find that the proposed MG can complement existing robust training methods and be easily extended to diverse advanced recommendation models, making it a promising new and fundamental paradigm for training multimodal recommender systems.

## CCS CONCEPTS

• **Information systems** → **Recommender systems**; • **Computing methodologies** → **Knowledge representation and reasoning**; **Artificial intelligence**.

## KEYWORDS

Recommender systems, Multimodal, Flat local minima, Robust

## 1 INTRODUCTION

**Relevance to the Web and to the track.** Recommender systems play a crucial role in helping users navigate the wealth of choices on the web and discover suitable items or online services. In fact, the integration of deep learning techniques into recommender systems has become widespread [4, 11, 41, 47]. These techniques leverage historical user-item interactions to model user preferences, thereby facilitating the personalized recommendation of items. In recent years, with the emergence of rich multimodal content information encompassing texts, images, and videos, multimodal recommender systems [33, 51] alleviate challenges [53] such as data sparsity and cold start. However, incorporating multimodal information into recommender systems also increases some inevitable risks about the input distribution shift.

The first risk is **inherent noise risk** which always appears in the training phase. Some previous works [38, 45] show that the performance of recommender systems encounters substantial challenges when confronted with input containing some noise in multimodal

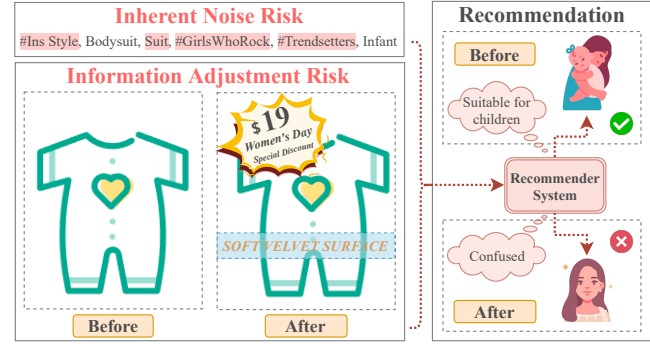

**Figure 1: An illustrative example of multimodal risks. Merchants add popular tags (e.g., "ins style") and broad keywords (e.g., "suit") to the text of the bodysuit to increase the likelihood of the item being recommended. At the same time, merchants dynamically change the item's visual features in real-time due to Women's Day marketing campaigns and the emphasis on the superiority of the item's material. These actions make it difficult for the recommender system to accurately determine the target user for the current item, leading to incorrect recommendations for young girls.**

information. These noises are intrinsic, such as subpar image quality of items or the presence of a significant number of irrelevant or error information in items' features. These factors contribute to inherent noise introduced to the model's input, and the introduction of multimodal data in multimodal recommender systems makes mitigating inherent noise risk more challenging. Another risk is **information adjustment risk**. After the recommender system has been trained based on multimodal data, it is well-known that multimodal data is prone to frequent adjustments. For example, merchants need to keep pace with trends or promotional activities to tailor keywords for items, and the descriptive images of items must be adjusted in line with iterative updates. This implies that in practical scenarios, the multimodal information within recommender systems often undergoes frequent modifications. A straightforward solution to address this risk is to update the model with the latest dataset, but as the volume of data increases, the cost of iterating the model significantly escalates, especially in multimodal recommender systems. In summary, these two risks pose a significant challenge [8, 27, 38, 45] to multimodal recommender systems. Fig. 1 is **an e-commerce case** to illustrate the effect of the risks. On one hand, during the training phase of recommender systems, inherent noise exists in multimodal features such as the text, which is intrinsic and has a potentially negative effect in the inference phase. On the other hand, during the inference phase of recommender systems, the multimodal feature as shown in Fig. 1 of the bodysuit is edited due to promotional activities and the emphasis on the

item's material. These two risks confuse the recommender system resulting in incorrect recommendations, which will be explicitly and quantitatively assessed in Section 6.

To mitigate the aforementioned risks of information adjustment and inherent noise, the necessity for building a more robust multimodal recommender system becomes apparent. For enhancing the robustness, prior efforts [8, 27] have chiefly employed adversarial training techniques. These methods improve the robustness of multimodal recommender systems by explicitly countering noise at input during the training phase. Different from them, in this paper, we first rethink the robustness of multimodal recommender systems from the loss landscape perspective by considering the *flat local minima* of multimodal recommendation models. Then, we propose a concise yet effective training strategy named Mirror Gradient (MG) for implicitly improving system robustness.

Specifically, we can first present an intuitive insight into why a recommender system should consider the flat local minima, which are located in large weight space regions with very similar low loss values [21]. In Fig. 2, $\ell_o$ represents the original loss landscape, which is associated with the model's parameters, architecture, data distribution, etc. When the system's inputs face shifts in data distribution due to risks like information adjustment or inherent noise, $\ell_o$ (transparent surface) also shifts to $\ell_s$ (opaque surface). If the model's parameters are optimized to a sharp local minimum $\theta_b$, the error caused by this shift $|\ell_s(\theta_b) - \ell_o(\theta_b)|$ may be significantly larger than $|\ell_s(\theta_a) - \ell_o(\theta_a)|$ of flat local minima $\theta_a$. This indicates that the system is not robust while the parameters are in sharp local minima. Therefore, we should strive to guide the learnable parameters of a multimodal recommender system towards flat local minima during training to enhance the model's robustness against potential risks of input distribution shifts.

To this end, we propose a concise gradient strategy MG that inverses the gradient signs appropriately during training to make the multimodal recommendation models approach flat local minima easier compared to models with normal training. Additionally, we conduct extensive experiments and analysis to validate the effectiveness of MG across various datasets and systems empirically. To elaborate our MG strategy, we first model it formally and then analyze how it improves the model's robustness by driving the parameters towards flat local minima implicitly from a theoretical perspective. The visualization method from Li et al. [25] supports our theoretical analysis and shows that MG indeed can help the model achieve flatter minima. Besides, we also empirically verify that MG is versatile, as it is compatible with most optimizers and other adversarial training-based robust recommendation methods.

In summary, our contributions are threefold:

- We analyze the robustness of multimodal recommender systems from the perspective of flat local minima.
- From the perspective of flat local minima, we propose Mirror Gradient (MG), a concise yet effective gradient strategy that guides recommender system models toward flat local minima, enhancing model robustness. We also provide theoretical evidence for its effectiveness.
- Extensive experiments demonstrate the efficacy and versatility of MG. We also discuss the limitations of MG.

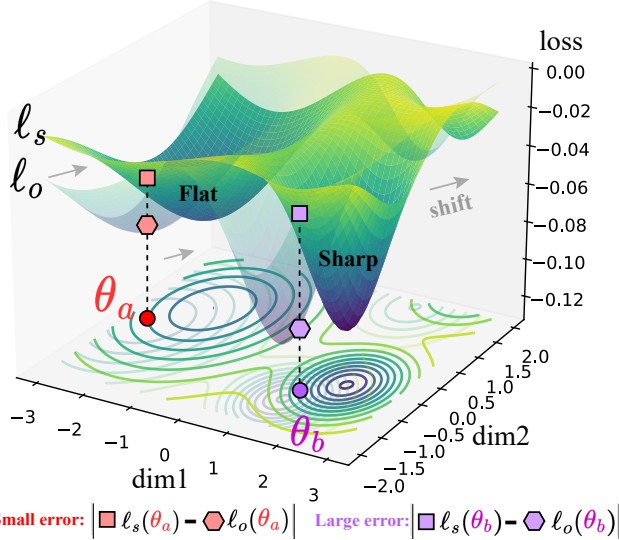

Small error: $\left| \square\, \ell_s(\theta_a) - \hexagon\, \ell_o(\theta_a) \right|$  Large error: $\left| \square\, \ell_s(\theta_b) - \hexagon\, \ell_o(\theta_b) \right|$

**Figure 2: Illustration of flat local minima. When the distribution of the inputs shifts, for example, facing the risks of inherent noise and information adjustment, the loss landscape $\ell_o$ of the recommender system also shifts (to $\ell_s$) accordingly. The parameters $\theta_a$ located in flat local minima are more robust compared to $\theta_b$ in sharp local minima.**

## 2 RELATED WORKS

**Multimodal Recommender Systems.** Traditional recommender systems [16, 48, 54] model the interaction between users and items, relying on extensive user-item interaction data to ensure accurate recommendations. In the presence of diverse multimodal information, multimodal recommender systems [1, 28, 31] leverage supplementary multimodal information to complement historical user-item interactions, mitigating challenges like data sparsity [53] and cold start [24, 34] in the recommendation. Early researchers often employed collaborative filter [41, 47] or matrix factorization [15] for multimodal recommendation modeling. Recently, many works [42, 52] employ graph neural networks for multimodal recommender systems, with self-supervised learning [40, 56] also gaining traction in this domain.

**Robust Recommender Systems.** Recent studies [5, 45] have shed light on the vulnerability of recommender systems, highlighting how disturbances introduced by noise can significantly undermine the accuracy of recommendations. In pursuit of bolstering the robustness of recommender systems, a multitude of efforts [3, 8, 27, 38] have focused on adversarial training. This approach, which operates under the premise that each instance may serve as a potential target for attacks [5], introduces controlled perturbations to either the input data or model parameters to enhance robustness. However, most existing studies have overlooked the potential risks arising from information adjustment in multimodal recommender systems.

**Flat Local Minima.** Flat local minima have been consistently linked to the favorable generalization capabilities of deep neural networks [14, 17–19, 35]. In the wake of this insight, numerous researchers have surfaced, exemplified by works such as Du

et al. [7], Zhao et al. [49], Zhuang et al. [57], which strive to enhance model performance through exploring flat local minima. Specifically, Foret et al. [10] introduces a novel approach named Sharpness-Aware Optimization (SAM), wherein the optimization process hinges on addressing a mini-max problem to achieve an optimal sharpness value. Kwon et al. [23], on the other hand, proposes a scale-invariant variant of SAM, named ASAM, bolstered by an adaptive radius mechanism aimed at augmenting training stability. Moreover, Mi et al. [30] innovatively delve into the realm of sparse perturbation with their SSAM (Sparse Sharpness-Aware Minima) approach, strategically focusing on perturbations that encapsulate the most critical yet sparse dimensions of the problem space.

# 3 PRELIMINARY

**Multimodal Recommender Systems.** Considering a set of users $\mathcal{U} = \{u_1, u_2, ..., u_{|\mathcal{U}|}\}$, and a set of items $\mathcal{I} = \{i_1, i_2, ..., i_{|\mathcal{I}|}\}$, each user $u \in \mathcal{U}$ is associated with an item set $\mathcal{I}_u \subseteq \mathcal{I}$ about which $u$ has expressed explicit positive feedback. Besides, each item $i \in \mathcal{I}$ has multimodal information denoted by visual features $v_i \in \mathcal{V}$ and textual features $t_i \in \mathcal{T}$ in this paper. Then given a multimodal recommendation model denoted as $\mathbf{R}(\cdot)$,

$$y_{u,i} = \mathbf{R}(u, i, v_i, t_i, \mathcal{I}_u \mid \Theta), \tag{1}$$

where $\Theta$ represents the model parameters of $\mathbf{R}(\cdot)$, and score $y_{u,i}$ signifies the preference of user $u$ towards item $i$. A higher score suggests that item $i$ is more suitable to be recommended to user $u$.
**Loss Function of Recommender Systems.** Most works [15, 50] optimize the model parameters $\Theta$ of multimodal recommender systems using Bayesian personalized ranking loss [32]. This optimization seeks to ensure that $y_{u,i}$, where $i \in \mathcal{I}_u$, is greater than $y_{u,i'}$ when $i' \notin \mathcal{I}_u$, thus promoting positive interactions while discouraging negatives ones. Additionally, some recommender systems introduce supplementary losses [40, 56] to enhance their performance. We adopt the unified notation $L_\mathrm{R}(\cdot)$ to represent those losses.

# 4 METHODOLOGY

In this section, we first elaborate on the algorithm of the proposed MG. Then, we introduce the theoretical insight of MG.

## 4.1 Mirror Gradient

MG is a concise and easily implementable approach that enhances the gradient of the model during the optimization process of recommender systems. This enhancement is equivalent to adding a regularization term to improve the system's robustness on input. The proposed MG consists of two phases in each training epoch: Normal Training and Mirror Training.

During Normal Training, the conventional gradient descent is applied to the loss function $L_\mathrm{R}(\cdot)$ with the current learnable parameters $\Theta_{t-1}$, as follows:

$$\Theta_t = \Theta_{t-1} - \eta \nabla_\Theta L_\mathrm{R}(\Theta_{t-1}), \tag{2}$$

where $\eta$ represents the learning rate. As shown in the Algorithm 1, we use an interval $\beta$ to control the effect of MG on each training epoch. After updating per $\beta - 1$ iterations using Eq. (2), we employ the Mirror Training strategy to update the parameter $\Theta_{t-1}$ as

---

**Algorithm 1** The training algorithm of Mirror Gradient

**Input:** The recommendation model $\mathbf{R}(\cdot)$; learning rate $\eta$; the number of iteration $T$; The scaling coefficients $\alpha_1, \alpha_2 \in \mathbb{R}^+$ and $\alpha_1 > \alpha_2$. The interval $\beta \in \mathbb{N}^+$ of mirror training.
**Output:** Model parameters $\Theta$.

1: **for** $t$ from 1 to $T$ **do**
2:      **if** $t \bmod \beta == 0$ **do**         ▷ Mirror Training
3:          $\Theta'_t = \Theta_{t-1} - \alpha_1 \eta \nabla_\Theta L_\mathrm{R}(\Theta_{t-1})$;
4:          $\Theta_t = \Theta'_t + \alpha_2 \eta \nabla_\Theta L_\mathrm{R}(\Theta'_t)$;
5:      **else do**                  ▷ Normal Training
6:          $\Theta_t = \Theta_{t-1} - \eta \nabla_\Theta L_\mathrm{R}(\Theta_{t-1})$;
7:      **end if**
8: **end for**
9: **return** $\Theta$

---

follows:

$$\begin{cases} \Theta'_t = \Theta_{t-1} - \alpha_1 \eta \nabla_\Theta L_\mathrm{R}(\Theta_{t-1}), \\ \Theta_t = \Theta'_t + \alpha_2 \eta \nabla_\Theta L_\mathrm{R}(\Theta'_t). \end{cases} \tag{3}$$

Here, in order to control the relative size of updates introduced by mirror training, we introduce two positive scaling coefficients, $\alpha_1$ and $\alpha_2$, with $\alpha_1 > \alpha_2$.

Although the MG we proposed is highly simple, it possesses a strong theoretical insight and remarkable versatility. This enables consistent performance improvements across a wide array of experimental scenarios. Furthermore, in Section 6, we demonstrate the compatibility of MG with various optimizers and existing robust recommender system techniques. Moreover, it can achieve superior performance compared to some conventional optimization strategies about flat local minima.

## 4.2 Theoretical Insight of MG

In this part, we introduce Lemma 4.1 and Theorem 4.2 which can help us analyze how MG helps the model's parameters tend towards flat local minima from a theoretical perspective, thereby enhancing the input robustness of the recommender system.

LEMMA 4.1. *[6, 20] Consider a neural network $f(x)$ with $L$ layers and learnable parameters $\theta$. $h_i, 1 \leq i \leq L$, denotes the feature map from $i$ th layer. For any scalar function $g$ of $h_L$, we have*

$$\|\nabla_x g_\theta(x)\|_2^2 \cdot \sum_{j=1}^L O\left(\frac{1 + \|h_i\|_2^2}{\|\nabla_x h_i\|_2^2}\right) \leq \|\nabla_\theta g_\theta(x)\|_2^2. \tag{4}$$

THEOREM 4.2. *Mirror Training in Eq. (3) is equal to introducing an implicit regularization term $\|\nabla_\Theta L_R(\Theta)\|_2^2$ to the original optimization objective $L_R(\Theta)$.*

PROOF. In Mirror Training, we have

$$\begin{aligned} \Theta_t &= \Theta'_t + \alpha_2 \eta \nabla_\Theta L_\mathrm{R}(\Theta'_t) \\ &= \Theta_{t-1} - \alpha_1 \eta \nabla_\Theta L_\mathrm{R}(\Theta_{t-1}) \\ &\quad + \alpha_2 \eta \nabla_\Theta L_\mathrm{R}(\Theta_{t-1} - \alpha_1 \eta \nabla_\Theta L_\mathrm{R}(\Theta_{t-1})). \end{aligned} \tag{5}$$

Next, since $\eta$ is small, we can use Taylor expansion for estimating $\nabla_\Theta L_R(\Theta_{t-1} - \alpha_1\eta\nabla_\Theta L_R(\Theta_{t-1}))$, and we have

$$
\begin{aligned}
\Theta_t &\approx \Theta_{t-1} - \alpha_1\eta\nabla_\Theta L_R(\Theta_{t-1}) + \alpha_2\eta\nabla_\Theta L_R(\Theta_{t-1}) \\
&\quad - \alpha_1\alpha_2\eta^2\nabla_\Theta^2 L_R(\Theta_{t-1})^\top\nabla_\Theta L_R(\Theta_{t-1}) \\
&= \Theta_{t-1} - (\alpha_1 - \alpha_2)\eta\nabla_\Theta L_R(\Theta_{t-1}) \\
&\quad - \frac{1}{2}\cdot\alpha_1\alpha_2\eta^2\nabla_\Theta\|\nabla_\Theta L_R(\Theta_{t-1})\|_2^2.
\end{aligned} \tag{6}
$$

Therefore, from Eq. (6), we can find that the equivalent objective function for Mirror Training $L_M$ is

$$
L_M = (\alpha_1 - \alpha_2)\underbrace{L_R(\Theta)}_{\text{main term}} + \alpha_1\alpha_2\eta\cdot\underbrace{\|\nabla_\Theta L_R(\Theta_{t-1})\|_2^2}_{\text{regularization term}}, \tag{7}
$$

where $\alpha_1\alpha_2\eta > 0$ and $\alpha_1 - \alpha_2 > 0$.  □

The essence of MG, as revealed in Theorem 4.2, lies in the addition of a regularization term concerning gradient magnitude to the original objective function $L_R(\Theta)$ implicitly. It's worth noting that the magnitude of gradients near flat local minima is quite small. And since $\alpha_1\alpha_2 > 0$, Eq. (6) requires that the norm of gradient $\|\nabla_\Theta L_R(\Theta)\|_2^2$ should be sufficiently small, i.e., MG will lead the model's parameters towards flatter minima. Furthermore, from Lemma 4.1 and Eq.(7), let the scalar function is $L_R$ in recommender system, we have

$$
\begin{aligned}
L_M &\geq \alpha_1\alpha_2\eta\cdot\|\nabla_\Theta L_R(\Theta_{t-1})\|_2^2 \\
&\geq \alpha_1\alpha_2\eta\cdot\sum_{j=1}^{L} O\left(\frac{1 + \|h_i\|_2^2}{\|\nabla_x h_i\|_2^2}\right)\cdot\underbrace{\|\nabla_x L_R\|_2^2}_{\text{robustness term}}.
\end{aligned} \tag{8}
$$

In general, $(1 + \|h_i\|_2^2)/(\|\nabla_x h_i\|_2^2)$ is bounded and positive. Taking BM3 on the Baby as an example, its value is around 96.16 with the well-trained system. Therefore, from Eq. (8) and Eq. (7), we can infer that MG also aims to minimize the impact of inputs on the loss, $\|\nabla_x L_R\|_2^2$, while enhancing the model's robustness against input perturbations.

Furthermore, although Eq. (7) reveals that our proposed MG is equivalent to adding a regularization term $\|\nabla_\Theta L_R(\Theta_{t-1})\|_2^2$ during the optimization process of recommender systems, we do not recommend directly including this term in the loss function. On one hand, computing this term requires the prior calculation of the gradient $\nabla_\Theta L_R(\Theta_{t-1})$, implying additional computational overhead during inference and not easy to implement. On the other hand, this kind of explicit loss term is not conducive to optimization and generally results in relatively poor performance [10]. In fact, our proposed MG is an implicit optimization of the additional regularization term shown in Eq. (7). It is also straightforward to implement. Hence, MG possesses greater practical applicability and potential.

## 5 EXPERIMENTS

In this section, we present our experimental setup and empirical results.

### 5.1 Experimental Settings

**Datasets.** The dataset statistics have been summarized in Table 1. We primarily employ four multimodal datasets from Amazon [29],

| Dataset | # Users | # Items | # Interactions | Sparsity |
|---|---|---|---|---|
| Pinterest | 3,226 | 4,998 | 9,844 | 99.94% |
| Baby | 19,445 | 7,050 | 160,792 | 99.88% |
| Sports | 35,598 | 18,357 | 296,337 | 99.95% |
| Clothing | 39,387 | 23,033 | 237,488 | 99.97% |
| Electronics | 192,403 | 63,001 | 1,689,188 | 99.99% |

Table 1: Statistics of datasets. These datasets comprise textual and visual features in the form of item descriptions and images.

| Model | REC | PREC | MAP | NDCG |
|---|---|---|---|---|
| VBPR | 0.0182 | 0.0042 | 0.0098 | 0.0122 |
| VBPR + MG | 0.0203 | 0.0046 | 0.0110 | 0.0136 |
| Improv. | 11.54% | 9.52% | 12.24% | 11.48% |
| MMGCN | 0.0140 | 0.0033 | 0.0075 | 0.0094 |
| MMGCN + MG | 0.0157 | 0.0036 | 0.0084 | 0.0106 |
| Improv. | 12.14% | 9.09% | 12.00% | 12.77% |
| GRCN | 0.0226 | 0.0051 | 0.0126 | 0.0155 |
| GRCN + MG | 0.0250 | 0.0057 | 0.0139 | 0.0171 |
| Improv. | 10.62% | 11.76% | 10.32% | 10.32% |
| DualGNN | 0.0238 | 0.0054 | 0.0132 | 0.0162 |
| DualGNN + MG | 0.0249 | 0.0056 | 0.0139 | 0.0170 |
| Improv. | 4.62% | 3.70% | 5.30% | 4.94% |
| BM3 | 0.0280 | 0.0062 | 0.0157 | 0.0192 |
| BM3 + MG | 0.0285 | 0.0063 | 0.0159 | 0.0195 |
| Improv. | 1.79% | 1.61% | 1.27% | 1.56% |
| FREEDOM | 0.0252 | 0.0056 | 0.0139 | 0.0171 |
| FREEDOM + MG | 0.0260 | 0.0058 | 0.0144 | 0.0176 |
| Improv. | 3.17% | 3.57% | 3.60% | 2.92% |
| **Avg. Improv.** | **7.31%** | **6.54%** | **7.46%** | **7.33%** |

Table 2: Top-5 recommendation performance of baselines with or without MG on Electronics. "Improv." indicates the relative enhancement of MG compared to the baseline. "Avg. Improv." represents the average improvement.

including Baby, Sports, Clothing, and Electronics. These datasets comprise textual and visual features in the form of item descriptions and images. Our data preprocessing methodology follows the approach outlined in Zhou et al. [51]. Furthermore, we utilize the dataset Pinterest to assess the compatibility of Mirror Gradient with adversarial training methods in alignment with the official implementation of the adversarial training baseline AMR [38].

**Metrics.** For the evaluation of recommendation performance, we pay attention to top-5 accuracy as recommendations in the top positions of rank lists are more important [39], and adopt four widely used metrics [13, 36, 46, 51] including recall (REC), precision (PREC), mean average precision (MAP), and normalized discounted cumulative gain (NDCG). These four evaluation metrics are chosen because they each focus on different crucial aspects. REC measures whether the system captures a user's potential areas of interest, PREC gauges the accuracy of recommendations, MAP focuses on the average accuracy of rankings, and NDCG emphasizes the quality

| Model | Baby | | | | Sports | | | | Clothing | | | |
|---|---|---|---|---|---|---|---|---|---|---|---|---|
| | REC | PREC | MAP | NDCG | REC | PREC | MAP | NDCG | REC | PREC | MAP | NDCG |
| VBPR | 0.0265 | 0.0059 | 0.0134 | 0.0170 | 0.0353 | 0.0079 | 0.0189 | 0.0235 | 0.0186 | 0.0039 | 0.0103 | 0.0124 |
| VBPR + MG | 0.0273 | 0.0061 | 0.0149 | 0.0184 | 0.0375 | 0.0084 | 0.0203 | 0.0251 | 0.0230 | 0.0048 | 0.0129 | 0.0155 |
| Improv. | 3.02% | 3.39% | 11.19% | 8.24% | 6.23% | 6.33% | 7.41% | 6.81% | 23.66% | 23.08% | 25.24% | 25.00% |
| MMGCN | 0.0240 | 0.0053 | 0.0130 | 0.0160 | 0.0216 | 0.0049 | 0.0114 | 0.0143 | 0.0130 | 0.0028 | 0.0073 | 0.0088 |
| MMGCN + MG | 0.0269 | 0.0060 | 0.0139 | 0.0175 | 0.0241 | 0.0054 | 0.0126 | 0.0158 | 0.0153 | 0.0032 | 0.0081 | 0.0100 |
| Improv. | 12.08% | 13.21% | 6.92% | 9.38% | 11.57% | 10.20% | 10.53% | 10.49% | 17.69% | 14.29% | 10.96% | 13.64% |
| GRCN | 0.0336 | 0.0074 | 0.0182 | 0.0225 | 0.0360 | 0.0080 | 0.0196 | 0.0241 | 0.0269 | 0.0056 | 0.0140 | 0.0173 |
| GRCN + MG | 0.0354 | 0.0078 | 0.0186 | 0.0232 | 0.0383 | 0.0086 | 0.0207 | 0.0256 | 0.0276 | 0.0058 | 0.0146 | 0.0179 |
| Improv. | 5.36% | 5.41% | 2.20% | 3.11% | 6.39% | 7.50% | 5.61% | 6.22% | 2.60% | 3.57% | 4.29% | 3.47% |
| DualGNN | 0.0322 | 0.0071 | 0.0175 | 0.0216 | 0.0374 | 0.0084 | 0.0206 | 0.0253 | 0.0277 | 0.0058 | 0.0153 | 0.0185 |
| DualGNN + MG | 0.0329 | 0.0073 | 0.0176 | 0.0219 | 0.0387 | 0.0086 | 0.0212 | 0.0261 | 0.0305 | 0.0063 | 0.0165 | 0.0201 |
| Improv. | 2.17% | 2.82% | 0.57% | 1.39% | 3.48% | 2.38% | 2.91% | 3.16% | 10.11% | 8.62% | 7.84% | 8.65% |
| SLMRec | 0.0343 | 0.0075 | 0.0182 | 0.0226 | 0.0429 | 0.0095 | 0.0233 | 0.0288 | 0.0292 | 0.0061 | 0.0163 | 0.0196 |
| SLMRec + MG | 0.0381 | 0.0085 | 0.0204 | 0.0253 | 0.0449 | 0.0099 | 0.0242 | 0.0299 | 0.0323 | 0.0067 | 0.0175 | 0.0213 |
| Improv. | 11.08% | 13.33% | 12.09% | 11.95% | 4.66% | 4.21% | 3.86% | 3.82% | 10.62% | 9.84% | 7.36% | 8.67% |
| BM3 | 0.0327 | 0.0072 | 0.0174 | 0.0216 | 0.0353 | 0.0078 | 0.0194 | 0.0238 | 0.0246 | 0.0051 | 0.0135 | 0.0164 |
| BM3 + MG | 0.0345 | 0.0077 | 0.0183 | 0.0228 | 0.0386 | 0.0086 | 0.0210 | 0.0259 | 0.0259 | 0.0054 | 0.0145 | 0.0174 |
| Improv. | 5.50% | 6.94% | 5.17% | 5.56% | 9.35% | 10.26% | 8.25% | 8.82% | 5.28% | 5.88% | 7.41% | 6.10% |
| FREEDOM | 0.0374 | 0.0083 | 0.0194 | 0.0243 | 0.0446 | 0.0098 | 0.0232 | 0.0291 | 0.0388 | 0.0080 | 0.0211 | 0.0257 |
| FREEDOM + MG | 0.0397 | 0.0088 | 0.0209 | 0.0261 | 0.0466 | 0.0102 | 0.0242 | 0.0303 | 0.0405 | 0.0084 | 0.0223 | 0.0270 |
| Improv. | 6.15% | 6.02% | 7.73% | 7.41% | 4.48% | 4.08% | 4.31% | 4.12% | 4.38% | 5.00% | 5.69% | 5.06% |
| DRAGON | 0.0374 | 0.0082 | 0.0202 | 0.0249 | 0.0449 | 0.0098 | 0.0239 | 0.0296 | 0.0401 | 0.0083 | 0.0225 | 0.0270 |
| DRAGON + MG | 0.0419 | 0.0092 | 0.0219 | 0.0273 | 0.0465 | 0.0102 | 0.0248 | 0.0307 | 0.0437 | 0.0091 | 0.0239 | 0.0290 |
| Improv. | 12.03% | 12.20% | 8.42% | 9.64% | 3.56% | 4.08% | 3.77% | 3.72% | 8.98% | 9.64% | 6.22% | 7.41% |
| **Avg. Improv.** † | **7.17%** | **7.91%** | **6.79%** | **7.08%** | **6.22%** | **6.13%** | **5.83%** | **5.90%** | **10.41%** | **9.99%** | **9.38%** | **9.75%** |

**Table 3: Top-5 recommendation performance of baselines with or without MG on Baby, Sport, and Clothing. "Improv." indicates the relative enhancement of MG compared to the baseline. "Avg. Improv." represents the average improvement across each dataset. † The Top-5 performance improvement achieved by MG is substantial and significant. See the appendix for further discussion.**

of rankings. These metrics complement each other and collectively provide a comprehensive evaluation, aiding in a holistic understanding of the recommender system's performance. Additionally, when comparing against the adversarial training method AMR [38], we apply the hits ratio (HR) in alignment with the evaluation approach used in the original AMR paper. The choice of HR serves the purpose of maintaining consistency and comparability with established benchmarks, allowing for a direct comparison of our results with those reported in the paper. All the above-cited metrics range from 0 to 1, the closer to 1 the better.

**Baselines.** We extensively examine MG's performance across a variety of multimodal recommendation models, encompassing matrix factorization (VBPR [15]), graph neural networks (MMGCN [44], GRCN [43], DualGNN [42], FREEDOM [52], DRAGON [50]), self-supervised learning (SLMRec [40], BM3 [56]), as well as non-multimodal models(LayerGCN [54], SelfCF [55]). We utilize AMR [38] as our foundational adversarial training method, given its widespread popularity in the field. Additionally, we compare our

MG with flat local minima approach SSAM [30], as it represents a leading-edge, versatile solution for addressing flat local minima.

**Implementation Details.** We retain the standard settings for all baselines. Following the settings of some current works in multimodal recommender systems [50, 52, 56], we perform a grid search on hyperparameters $\alpha_1$ and $\alpha_2$, and set $\beta$ to 3. Unless otherwise specified, Adam [22] serves as the chosen optimizer. The training and evaluation of all models is conducted using the RTX3090 GPU.

## 5.2 Overall Performance

**Observation #1: MG can enhance the performance of diverse multimodal recommender systems consistently.** As shown in Table 3, we conduct an extensive evaluation of MG across eight baseline models using three distinct datasets. Our experimental findings unequivocally illustrate that it is difficult to overlook the enhancements in the performance of multimodal recommender systems with MG across all evaluation metrics. Remarkably, the most substantial improvement is observed in the case of VBPR training on the dataset Clothing, resulting in an impressive performance enhancement of over 20%. To summarize, MG we propose excels at

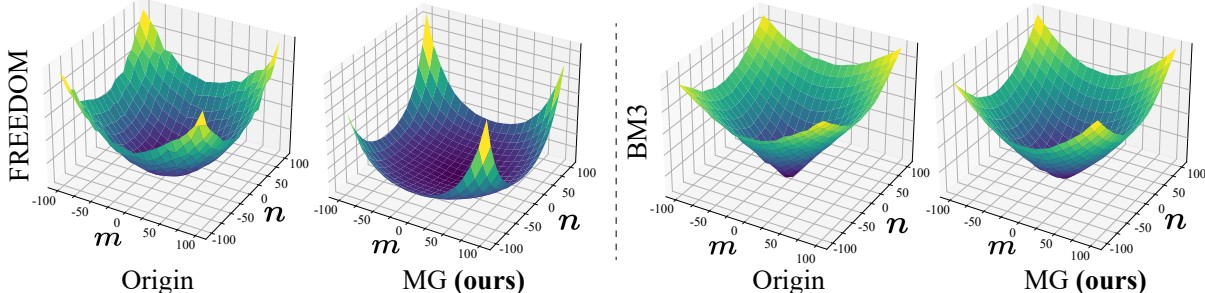

Figure 3: Visualization of local minima. Training loss landscapes of FREEDOM and BM3 on Baby trained with or without MG.

mitigating inherent noise risks, enabling multimodal recommender systems to capture user preferences within more favorable flat local minima. This capability contributes to the success of our method.

**Observation #2: MG exhibits remarkable efficacy on more challenging datasets.** Furthermore, as indicated in Table 2, we extend the evaluation of our proposed MG to the dataset Electronics, which presents a more demanding scenario with a cumulative user and item count surpassing the sum of other datasets presented in Table 3. Remarkably, the increased challenge posed by this dataset does not diminish MG's performance. On Electronics, the average improvement achieved by MG remains consistent with the figures presented in Table 3.

## 6 ANALYSIS

Our analysis is designed to answer the following research questions (RQs), where RQ1-5 focus on the performance of MG, while RQ6-8 examine its versatility:

- RQ1: How does MG perform in mitigating inherent noise risk?
- RQ2: How does MG perform in mitigating information adjustment risk?
- RQ3: Does MG enable multimodal recommender systems to approach flatter local minima?
- RQ4: Does MG increase model training costs?
- RQ5: Is MG superior to sharpness-aware minimization methods?
- RQ6: Can MG be compatible with various optimizers?
- RQ7: Can MG be compatible with robust recommender systems?
- RQ8: Can MG enhance the performance of non-multimodal recommender systems?

**Further Evaluation for MG's Robustness (RQ1 & RQ2).** In this part, we directly validate the robustness of the proposed MG by explicitly simulating the two risks mentioned in Section 1 in the inference phase.

(1) Information noise. For models with learnable item embedding layers, e.g., BM3 and FREEDOM, we inject Gaussian noise $\epsilon \sim \mathcal{N}(0, 10^{-6})$ into their embedding layers. This step aims to simulate the presence of information noise in multimodal recommender systems. The results are presented in Table 4.

(2) Information adjustment. To simulate the adjustment in multimodal information within a recommender system, we insert image captions generated using BLIP-2 [26] into the text of the dataset with a one percent chance. As models with learnable item embedding layers do not rely on the original multimodal information

| Model | Org | Noise | Decr. ↓ |
|---|---|---|---|
| BM3 | 0.0327 | 0.0272 | 16.92% |
| BM3 + MG | 0.0345 | 0.0314 | **8.99%** |
| FREEDOM | 0.0374 | 0.0357 | 4.60% |
| FREEDOM + MG | 0.0397 | 0.0390 | **1.76%** |

Table 4: REC of models under information noise on Baby. The term "Noise" refers to the injection with noise $\epsilon \sim \mathcal{N}(0, 10^{-6})$ into the embedding layers of the model. The experimental results of the noise are calculated as the mean value obtained from conducting the noise experiment 10 times repetitively. "Decr." denotes the relative decrease compared to the origin result (Org) after injecting noise.

| Model | Org | Adjustment | Decr. ↓ |
|---|---|---|---|
| MMGCN | 0.0240 | 0.0223 | 7.08% |
| MMGCN + MG | 0.0269 | 0.0260 | **3.35%** |
| GRCN | 0.0336 | 0.0321 | 4.46% |
| GRCN + MG | 0.0354 | 0.0346 | **2.26%** |

Table 5: REC of models under information adjustment on Baby. The term "Adjustment" refers to the action of inserting image captions generated using BLIP-2 [26] into the text of items with a 1% chance. This insertion process simulates changes in multimodal information within recommender systems. "Decr." denotes the relative decrease compared to the origin result (Org) after information adjustment.

during the inference phase, we select MMGCN and GRCN, which use deep neural networks to extract multimodal information, as the baselines. The results are outlined in Table 5.

The results of experiments indicate that multimodal recommendation models trained with MG exhibit greater robustness in handling multimodal noise and adjustment.

**Visualization of Local Minima (RQ3).** In order to affirm that MG can help the model approach flat local minima, we visualize the training loss landscapes with and without MG of latest models BM3 and FREEDOM which is more challenge to improve than previous models. Following the methodology outlined by Li et al. [25], we record the trained model parameters as $p$. Subsequently, we sample 20 values each for $m$ and $n$ at equal intervals from the

| Optimizer | Model | REC | PREC | MAP | NDCG | Model | REC | PREC | MAP | NDCG |
|---|---|---|---|---|---|---|---|---|---|---|
| Adam | GRCN | 0.0336 | 0.0074 | 0.0182 | 0.0225 | DRAGON | 0.0374 | 0.0082 | 0.0202 | 0.0249 |
| | GRCN + MG | 0.0354 | 0.0078 | 0.0186 | 0.0232 | DRAGON + MG | 0.0419 | 0.0092 | 0.0219 | 0.0273 |
| | Improv. | 5.36% | 5.41% | 2.20% | 3.11% | Improv. | 12.03% | 12.20% | 8.42% | 9.64% |
| SGD | GRCN | 0.0013 | 0.0003 | 0.0005 | 0.0007 | DRAGON | 0.0182 | 0.0040 | 0.0093 | 0.0118 |
| | GRCN + MG | 0.0014 | 0.0003 | 0.0006 | 0.0008 | DRAGON + MG | 0.0188 | 0.0041 | 0.0097 | 0.0122 |
| | Improv. | 7.69% | 0.00% | 20.00% | 14.29% | Improv. | 3.30% | 2.50% | 4.30% | 3.39% |
| RMSprop | GRCN | 0.0338 | 0.0074 | 0.0184 | 0.0227 | DRAGON | 0.0367 | 0.0081 | 0.0198 | 0.0245 |
| | GRCN + MG | 0.0345 | 0.0076 | 0.0187 | 0.0231 | DRAGON + MG | 0.0391 | 0.0087 | 0.0201 | 0.0253 |
| | Improv. | 2.03% | 2.63% | 1.60% | 1.73% | Improv. | 6.54% | 7.41% | 1.52% | 3.27% |
| Adagrad | GRCN | 0.0283 | 0.0063 | 0.0152 | 0.0189 | DRAGON | 0.0393 | 0.0086 | 0.0215 | 0.0264 |
| | GRCN + MG | 0.0286 | 0.0064 | 0.0154 | 0.0191 | DRAGON + MG | 0.0408 | 0.0090 | 0.0216 | 0.0269 |
| | Improv. | 1.06% | 1.59% | 1.32% | 1.06% | Improv. | 3.82% | 4.65% | 0.47% | 1.89% |

**Table 6: Top-5 recommendation performance of GRCN and DRAGON with or without MG on Baby. "Improv." indicates the relative enhancement of MG compared to the baselines.**

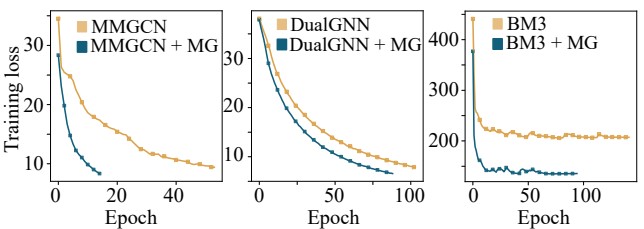

**Figure 4: Convergence of MG on the dataset Baby.**

| Method | REC | PREC | MAP | NDCG |
|---|---|---|---|---|
| - | 0.0374 | 0.0082 | 0.0202 | 0.0249 |
| SSAM-F | 0.0397 | 0.0088 | 0.0217 | 0.0267 |
| SSAM-D | 0.0382 | 0.0084 | 0.0197 | 0.0248 |
| MG | 0.0419 | 0.0092 | 0.0219 | 0.0273 |

**Table 7: Top-5 recommendation performance of DRAGON with flat local minima method SSAM and MG on Baby. SSAM-F draws upon fisher information and SSAM-D capitalizes on the principles of dynamic sparse training mask.**

range $[-100, 100]$, and init noise $n_1, n_2$ from the standard normal distribution. $n_1, n_2$ have the same shape as $p$. We then update the model's parameters to $(p + mn_1 + nn_2)$ and calculate the corresponding loss values, simulating the shift of the loss landscape as depicted in Fig. 2. By employing this methodology, we can generate training loss landscapes as illustrated in Fig. 3. The flatter the training loss landscape, the flatter the local minima to which the current model converges. Notably, the landscapes associated with MG demonstrate a flatter topography when compared to those of the baseline approaches and prominently encompass an area characterized by low loss (depicted in blue). These visualization results show that MG can make multimodal recommender system approach flatter minima. Moreover, from the visualization results, the loss landscape associated with FREEDOM appears flatter compared to that of BM3. This indicates that FREEDOM may be more robust to noise, aligning with the observations in Table 4.

**Convergence Speed of MG (RQ4).** In this part, we visualize the training loss of both widely used and leading-stage models to confirm MG's superior convergence. Following the training configuration outlined by Zhou et al. [51], we set the maximum number of epochs to 1000 while implementing an early stopping strategy. Subsequently, we visualize the progression of the loss value for multiple multimodal recommendation models before and after the application of MG, as shown in Fig. 4. Noticeably, MG often leads the models to meet the termination criteria with fewer training iterations and achieve a smaller final training loss value.

**Comparing with Sharpness-aware Minimization (RQ5).** Recently, several general smoothing methods [7, 14, 35, 49, 57] about flat local minima have been proposed. In this section, we compare these methods with MG in multimodal recommender systems. Specifically, we apply the recent method SSAM [30] to the multimodal recommendation model DRAGON on the dataset Baby, and the results are laid out in Table 7. The experimental results show that MG outperforms both SSAM-F, which draws upon fisher information, and SSAM-D, which capitalizes on the principles of dynamic sparse training mask [30]. These experiments suggest the MG's inherent advantage of identifying flat local minima compared with other minimization methods in the scenario of multimodal recommender systems.

**Compatibility with Various Optimizers (RQ6).** As a gradient method, it is necessary for MG to adapt to various optimizers. Therefore, in this part, we evaluate the performance of MG on the dataset Baby under various optimizers and baselines as illustrated in Table 6. The optimizers include Adam [22], SGD [2, 37], RMSprop [12], and Adagrad [9]. Here, we choose the classic multimodal recommender system GRCN and the latest state-of-the-art multimodal recommender system DRAGON as baselines. Despite the considerable performance fluctuations observed among baselines under different optimizers, MG consistently delivers a noticeable enhancement in recommendation accuracy. This consistency highlights the stable performance of MG across a range of optimizers.

| Model | HR | NDCG |
|---|---|---|
| VBPR | 0.1352 | 0.1005 |
| VBPR + AMR | 0.1395 | 0.1027 |
| VBPR + AMR + MG | 0.1457 | 0.1047 |

**Table 8: Top-5 recommendation performance of VBPR with adversarial training method AMR and MG on Pinterest.**

| Dataset | Metric | LayerGCN | LayerGCN + MG | SelfCF | SelfCF + MG |
|---|---|---|---|---|---|
| Baby | REC | 0.0314 | 0.0337 | 0.0329 | 0.0348 |
| | PREC | 0.0070 | 0.0075 | 0.0073 | 0.0078 |
| | MAP | 0.0168 | 0.0179 | 0.0175 | 0.0182 |
| | NDCG | 0.0209 | 0.0223 | 0.0217 | 0.0228 |
| Clothing | REC | 0.0242 | 0.0258 | 0.0246 | 0.0284 |
| | PREC | 0.0051 | 0.0054 | 0.0052 | 0.0059 |
| | MAP | 0.0136 | 0.0141 | 0.0136 | 0.0155 |
| | NDCG | 0.0163 | 0.0171 | 0.0164 | 0.0188 |

**Table 9: Top-5 recommendation performance of general models with or without MG.**

**Compatibility with Other Robust Recommenderation Methods (RQ7).** Researchers always enhance the robustness of multimodal recommender systems in the way of adversarial training [3, 8, 27, 38]. Therefore, in this part, we perform experiments using the official code of adversarial training method AMR [38] to explore the performance of multimodal recommender systems training with both MG and the robust recommendation method. As presented in Table 8, the experimental results show that the performance of VBPR with AMR can be improved by MG, indicating that MG is compatible with adversarial training aiming at enhancing the robustness of multimodal recommender systems.

**MG for Non-multimodal Systems (RQ8).** The non-multimodal system is a special case of multimodal settings. In this part, we validate the effectiveness of MG across various classical non-multimodal recommender systems, like LightGCN [54] and SelfCF [55]. Although they are subjected to a relatively lower risk of input distribution shift, as indicated in Table 9, our proposed MG also brings a noticeable improvement in the performance of these general models, showing that even in a single-modal context, MG remains effective in identifying superior flat local minima and promoting model robustness.

## 7 ABLATION STUDY

**The Interval $\beta$ in Algorithm 1.** $\beta$ denotes the interval of mirror training, where MG affects the training of multimodal recommendation models every $\beta$ iterations. As depicted in Table 10, we set $\beta$ to 1, 3, 5, and 7, and observe the performance of various multimodal recommendation models. Interestingly, we notice that changes in $\beta$ do not lead to a significant impact on model performance, with 3 being a suitable choice for the value of $\beta$.

**The $(\alpha_1, \alpha_2)$ in Algorithm 1.** $\alpha_1$ and $\alpha_2$ are the scaling coefficients while using Mirror Training. As shown in Fig. 5, when $\alpha_1 > \alpha_2$, MG can achieve significant performance improvement by setting appropriate $(\alpha_1, \alpha_2)$. From our theoretical results in Eq. (7), when $\alpha_1 = \alpha_2$ (red dashed line in Fig. 5), the objective function degenerates to

| Model | Metric | $\beta = 1$ | $\beta = 3$ | $\beta = 5$ | $\beta = 7$ |
|---|---|---|---|---|---|
| DualGNN | REC | 0.0327 | 0.0329 | **0.0331** | 0.0327 |
| | NDCG | 0.0218 | **0.0219** | 0.0217 | 0.0216 |
| BM3 | REC | 0.0339 | **0.0345** | 0.0337 | 0.0342 |
| | NDCG | 0.0223 | **0.0228** | 0.0223 | 0.0225 |
| DRAGON | REC | 0.0418 | **0.0419** | 0.0410 | 0.0416 |
| | NDCG | **0.0274** | 0.0273 | 0.0273 | **0.0274** |

**Table 10: Top-5 recommendation performance of models with different MG interval ($\beta$) on Baby.**

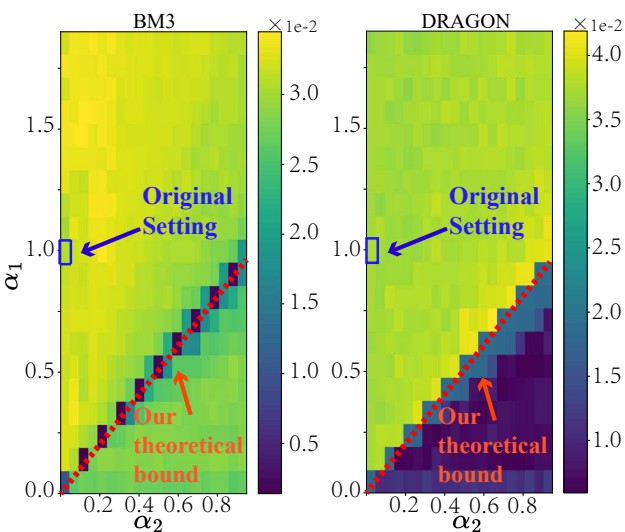

**Figure 5: Recall for different $(\alpha_1, \alpha_2)$ on Baby.**

only the regularization term, leading to optimization difficulties. On the other hand, when $\alpha_1 < \alpha_2$ (below the red dashed line), the main term becomes a pure gradient ascent term, which might not necessarily be advantageous for optimization and potentially impact the model's performance. These observations align well with our theoretical results.

## 8 LIMITATIONS

The selection of $(\alpha_1, \alpha_2)$ lacks a direct and efficient method, often requiring grid search, similar to other advanced recommender system approaches [50, 52, 56]. Moreover, as per Algorithm 1, MG theoretically requires more time for iterations. Fortunately, as indicated by the results in Fig. 4, MG exhibits rapid convergence speed, making the additional computational cost acceptable. In the future, further development of MG is needed to address these limitations.

## 9 CONCLUSION

In this paper, we rethink the robustness challenges in multimodal recommender systems through the perspective of flat local minima. We propose a novel gradient strategy called Mirror Gradient (MG) to address the prevalent risk of input distribution shift in the systems. Extensive experiments across diverse multimodal recommendation models and benchmark datasets and strong theoretical evidence demonstrate the effectiveness, compatibility, and versatility of MG.

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

# A SUPPLEMENTARY EXPERIMENTS

Throughout all the experiments in this paper, we adopted the Top-5 recommendation performance as the metric due to its capacity to reflect user preferences better, given the heightened significance of recommendations in top positions of rank lists [39]. However, enhancing Top-5 performance is a challenging job on Amazon datasets. The current advanced multimodal recommender system methods, including matrix factorization (VBPR [15]), self-supervised learning (SLMRec [40], BM3 [56]), graph neural networks (MMGCN [44], GRCN [43], DualGNN [42], FREEDOM [52], DRAGON [50]), still achieve relatively low value on Top-5 recommendation performance.

In fact, despite the challenge in enhancing this metric, we believe that the improvement brought about by the proposed MG in this paper is substantial and significant. Specifically, in Table 11 and Table 12, we present the performance evaluation of the BM3 and FREEDOM methods on the dataset Baby under various random seeds. Compared to the performance of the original method under different random seeds, the improvement achieved by MG is notable. This implies that the enhancements from MG are substantial and not obtained through random perturbations or simple parameter-tuning.

| BM3 | REC | Improv. | PREC | Improv. | MAP | Improv. | NDCG | Improv. |
|---|---|---|---|---|---|---|---|---|
| Origin | 0.0327 | 0.00% | 0.0072 | 0.00% | 0.0174 | 0.00% | 0.0216 | 0.00% |
| | 0.0329 | 0.61% | 0.0073 | 1.39% | 0.0176 | 1.15% | 0.0218 | 0.93% |
| | 0.0329 | 0.61% | 0.0073 | 1.39% | 0.0176 | 1.15% | 0.0218 | 0.93% |
| | 0.0329 | 0.61% | 0.0073 | 1.39% | 0.0173 | -0.57% | 0.0216 | 0.00% |
| | 0.0323 | -1.22% | 0.0072 | 0.00% | 0.0175 | 0.57% | 0.0216 | 0.00% |
| Random seed | 0.0323 | -1.22% | 0.0072 | 0.00% | 0.0174 | 0.00% | 0.0216 | 0.00% |
| | 0.0322 | -1.53% | 0.0072 | 0.00% | 0.0174 | 0.00% | 0.0215 | -0.46% |
| | 0.0322 | -1.53% | 0.0072 | 0.00% | 0.0174 | 0.00% | 0.0215 | -0.46% |
| | 0.0321 | -1.83% | 0.0072 | 0.00% | 0.0174 | 0.00% | 0.0215 | -0.46% |
| | 0.0321 | -1.83% | 0.0072 | 0.00% | 0.0174 | 0.00% | 0.0215 | -0.46% |
| MG | 0.0345 | 5.50% | 0.0077 | 6.94% | 0.0183 | 5.17% | 0.0228 | 5.56% |

Table 11: Top-5 recommendation performance of BM3 on the dataset Baby. "Random seed" denotes that the model is trained with a random seed.

| FREEDOM | REC | Improv. | PREC | Improv. | MAP | Improv. | NDCG | Improv. |
|---|---|---|---|---|---|---|---|---|
| Origin | 0.0374 | 0.00% | 0.0083 | 0.00% | 0.0194 | 0.00% | 0.0243 | 0.00% |
| | 0.0376 | 0.53% | 0.0083 | 0.00% | 0.0193 | -0.52% | 0.0243 | 0.00% |
| | 0.0376 | 0.53% | 0.0082 | -1.20% | 0.0195 | 0.52% | 0.0246 | 1.23% |
| | 0.0374 | 0.00% | 0.0082 | -1.20% | 0.0194 | 0.00% | 0.0245 | 0.82% |
| | 0.0374 | 0.00% | 0.0083 | 0.00% | 0.0195 | 0.52% | 0.0246 | 1.23% |
| Random seed | 0.0373 | -0.27% | 0.0082 | -1.20% | 0.0196 | 1.03% | 0.0244 | 0.41% |
| | 0.0373 | -0.27% | 0.0082 | -1.20% | 0.0196 | 1.03% | 0.0245 | 0.82% |
| | 0.0373 | -0.27% | 0.0082 | -1.20% | 0.0196 | 1.03% | 0.0245 | 0.82% |
| | 0.0373 | -0.27% | 0.0082 | -1.20% | 0.0196 | 1.03% | 0.0245 | 0.82% |
| | 0.0372 | -0.53% | 0.0082 | -1.20% | 0.0196 | 1.03% | 0.0245 | 0.82% |
| MG | 0.0397 | 6.15% | 0.0088 | 6.02% | 0.0209 | 7.73% | 0.0261 | 7.41% |

Table 12: Top-5 recommendation performance of FREEDOM on the dataset Baby. "Random seed" denotes that the model is trained with a random seed.

