# OpenReview forum: "Mirror Gradient: Towards Robust Multimodal Recommender Systems via Exploring Flat Local Minima"
_ACM.org/TheWebConf/2024/Conference — TheWebConf24_

### Official Review · Reviewer_8CsH · 2023-11-23

**Novelty:** 4
**Technical Quality:** 3

**Review:**

### Pros
1. The idea of approximating flat local minima by alternative updating parameters with different learning ratios is quite interesting. It regularizes the second-order derivation with two first-order gradients.
2. The theoretical bound is intuitive, and the proof is easy to follow.

### Cons
1. Why recommender system? It appears the paper has a limited relationship to the web. It is more like a machine learning paper, with its experiment conducted only on the recommender system.
2. Why focus on the multi-modality? It appears the method can work on any robust ML task.
3. No statistical analysis is conducted.

**Questions:**

See the cons

**Reviewer Confidence:**

3: The reviewer is confident but not certain that the evaluation is correct

**Scope:**

3: The work is somewhat relevant to the Web and to the track, and is of narrow interest to a sub-community

---

### Official Review · Reviewer_bird · 2023-11-24

**Novelty:** 6
**Technical Quality:** 6

**Review:**

This paper has proposed a novel gradient strategy, Mirror Gradient (MG), aimed at improving the robustness of multimodal recommender systems during the optimization process. This strategy is designed to mitigate the instability risks that arise from multimodal information inputs. Both theoretical analysis and experimental results validated the superiority of the proposed methods.


### Pros
1. The analysis of robustness in multimodal recommender systems from the perspective of flat local minima exhibits a certain degree of novelty.
2. The authors provide strong theoretical backing for the proposed MG and validate its effectiveness through extensive empirical experiments across various multimodal recommendation models and benchmarks.
3. The authors demonstrate that MG can supplement existing robust training methods and has the potential to be extended to a wide array of advanced recommendation models. This positions MG as a potentially groundbreaking approach in the training of multimodal recommender systems.
4. The paper is well-organized and presents a comprehensive list of research questions.

### Cons
1. It is suggested to carry out experiments concerning information noise/adjustment under varying levels of noise/adjustment, as detailed in Tables 4 and 5.
2. It is recommended to place a conclusion next to the figure caption (Fig. 3).
3. Miscellaneous issues: On line 381, BM3 is mentioned for the first time and should be properly referenced. Line 580 needs to be revised.

**Questions:**

How about the top-10 performance of MG? For example, on Baby?

**Reviewer Confidence:**

4: The reviewer is certain that the evaluation is correct and very familiar with the relevant literature

**Scope:**

4: The work is relevant to the Web and to the track, and is of broad interest to the community

---

### Official Review · Reviewer_Yfas · 2023-11-26

**Novelty:** 5
**Technical Quality:** 6

**Review:**

This paper presents an approach to improving the robustness of multimodal recommender systems by introducing a gradient strategy called Mirror Gradient (MG).
Pros:
1. The authors provide both theoretical evidence and empirical results to support their claims. This combination strengthens the validity of their findings.

2. The authors claim that the proposed MG can be easily extended to diverse advanced recommendation models, which suggests a broad applicability of their work.

3. The proposed method is easy to implement and be applied to other models.

Cons:
1. The paper could benefit from a more detailed explanation of the proposed MG strategy.

2. The paper could benefit from a more detailed comparison with other methods. This would help readers understand the unique advantages of the proposed MG strategy.

**Questions:**

1. I wonder If your Mirror Gradient technique is related to or inspired by previous method. it would be beneficial to clarify this in Section Related work of your paper and discuss how your method differs or improves upon existing flat local minima methods and robust recommender systems.

2. Since this paper mentioned that MG is equivalent to adding a new regularization term, it is important to compare MG with other existing regularization techniques (such as L1, L2 regularization) to demonstrate its advantages and potential benefits.

3. In line 349, the author mentions that when 𝜂 is small, it can be approximated using a Taylor expansion. For the benefit of readers, I recommend providing a more detailed explanation or justification.

4. Since the method is simple to implement, providing implementation details with PyTorch or TensorFlow code in this paper is indeed a great way to enhance reproducibility and aid reader comprehension.

5. What’s the performance compared with other flat local minima methods? Such as ASAM SAM, mentioned in related works.

6. In the section where authors compare baselines with and without MG, I noticed that you adjusted the hyperparameters 𝛼1 and 𝛼2 and set 𝛽 to 3. I am curious to know if other hyperparameters and settings of baselines were kept the same?

7. A block may need to be deleted in Line 363.

**Reviewer Confidence:**

3: The reviewer is confident but not certain that the evaluation is correct

**Scope:**

4: The work is relevant to the Web and to the track, and is of broad interest to the community

---

### Official Review · Reviewer_eXt9 · 2023-11-28

**Novelty:** 4
**Technical Quality:** 3

**Review:**

This paper discusses a novel approach in the field of multimodal recommender systems, focusing on enhancing the robustness against inherent risks like information adjustment and inherent noise. To achieve this, this paper introduces the Mirror Gradient (MG) method, which is designed to implicitly guide the model's parameters towards flat local minima during the optimization process. To be specific, MG operates in two phases during each training epoch - Normal Training and Mirror Training. And the authors claim that  MG is
equivalent to adding a regularization term in a form of gradient normalization. The effectiveness of MG is validated through extensive experiments and analysis across various datasets and systems.

Strengths
Innovative Approach: MG introduces a novel perspective in handling the robustness of multimodal recommender systems, focusing on guiding parameters towards flat local minima, a relatively unexplored area.
Compatibility and Versatility: MG is shown to be compatible with most optimizers and can complement existing robust training methods, indicating its potential for wide applicability in diverse recommender systems.

Weaknesses
Theoretical Justification: The paper provides theoretical evidence for the insight of MG. However, the application of Taylor expansion is confusing. It is not clear at which value the given function is expanded and how to get the next expression.
Empirical Validation: The authors claim that MG considers the flat local minima, but in experiments it does not compare to other flat local minima approaches.

In conclusion, it seems that there are still some unclear problems in this paper, and thus it is not ready for publication in the Web
Conference.

**Questions:**

1. MG only modifies the process of parameter optimization and does not change the loss function directly. How does MG affect the loss function landscape since loss function does not change?
2. The authors claim that MG is equivalent to adding a regularization term in a form of gradient normalization. In that case, why not just add a gradient norm to the loss function like Zhao et al. [49]?
3. In experiments, why not compare other flat local minima approaches?

**Reviewer Confidence:**

3: The reviewer is confident but not certain that the evaluation is correct

**Scope:**

4: The work is relevant to the Web and to the track, and is of broad interest to the community

---

### Decision · Program_Chairs · 2024-01-22

**Decision:**

Accept

**Comment:**

This paper looks at improving the robustness of multimodal recommender systems. Specifically, the authors propose a new training approach that aims to improve the optimization process by guiding model parameters towards flat local minima, claiming this is equivalent to adding a regularization term, which is infeasible to do directly.

 Key strengths:
 * The approach of helping identify flat local minimal is novel, and appears to be applicable to most optimization approaches thus complementing existing literature.
 * The authors present both theoretical and empirical justifications for their approach.
 * The method is relatively simple to implement, helping reproducibility and future research building on this work.

 Key weaknesses:
 * The writing is undoubtedly confusing. Some of the reviewers were left with significant questions. It is critical that these are addressed for future readers to be able to understand the paper and its claims.
 * More detailed comparison to other work would be very helpful.

 Discussion:
 * The authors provided very extensive responses to all the reviewer questions.
 * The authors performed many new experiments, with these constituting new work rather than clarifications. However, these do not change the conclusions and instead answer reviewer questions. The paper likely won't have space in the main body for much of this to be included in the camera ready version.
 * The authors also provided more detailed derivations of the more confusing theoretical aspects. These are helpful in building a better understanding of what is happening, and can be incorporated into the camera ready version easily.
 * The authors have promised to release their code.

 Summary:
 While somewhat confusing in it's presentation, the core of the paper's contribution appears to be solid and backed up by strong theoretical and empirical evidence.